# Acceleration of Aerobic Granulation in Sidestream Treatment with Exogenous Autoinducer

Eunae Jang [1,†], Kyung Jin Min [2,†], Eunyoung Lee [1], Hanna Choi [3] and Ki Young Park [1,*]

1. Department of Civil and Environmental Engineering, Konkuk University, 120 Neungdong-ro, Gwangjin-gu, Seoul 05029, Republic of Korea; eunae613@konkuk.ac.kr (E.J.); eylee84@gmail.com (E.L.)
2. Department of Tech Center for Research Facilities, Konkuk University, 120 Neungdong-ro, Gwangjin-gu, Seoul 05029, Republic of Korea; kyungjinm@konkuk.ac.kr
3. Taeyoung E&C, 111 Yeouigongwon-ro, Yeongdeungpo-gu, Seoul 07241, Republic of Korea; hnchoi@taeyoung.com
* Correspondence: kypark@konkuk.ac.kr; Tel.: +82-2-447-3637
† These authors contributed equally to this work.

**Abstract:** Aerobic granular sludge (AGS) is a special type of biofilm formed by the self-aggregation of microorganisms and extracellular polymers and is considered a promising technology for wastewater treatment. However, new strategies are still being proposed as to how to improve the extracellular polymeric substances (EPS) production that influences the formation of AGS. Recently, the acceleration of aerobic granulation using autoinducers such as N-acyl-homoserine lactone (AHL)-mediated quorum sensing has been reported. However, it is not yet fully understood due to knowledge gaps on the correlations depending on the type of AHL used. In this study, to evaluate the effects of various AHL on the AGS formation of activated sludge, the secretion of extracellular polymeric substances, biofilm formation, and sludge characteristics were comprehensively investigated. Among the AHL types, tightly bound EPS (TB-EPS) and loosely bound EPS (LB-EPS) in the reactor with C8-HSL added were 18.49 and 74.07 mg/g VSS, respectively, which represented increases of 3.15% and 53.76% compared to the control group. Additionally, C8-HSL increased the relative hydrophobicity and biomass volume by 153% and 218%, respectively. As a result, AHL had a positive effect on biomass content, an increase in sludge size, and an improvement in sludge sedimentation in the early stage of granulation, and C8-HSL was found to be the most suitable for initial granulation among AHL types.

**Keywords:** aerobic granular sludge; autoinducer; quorum sensing; *N*-acyl-homoserine lactone; extracellular polymeric substance; hydrophobicity

## 1. Introduction

Aerobic granular sludge (AGS) is a special biofilm formed through the self-fixation of microorganisms under aerobic conditions and is considered a promising technology for water treatment [1]. AGS has been extensively studied at various scales over the past few decades because of its advantages, such as stable sedimentation, securing high-concentration biomass, shortening residence time in wastewater treatment, and maintaining high treatment performance for high-concentration organic wastewater [2]. Because AGS is generally formed in a reactor with a large height/diameter ratio, the vertical space can be efficiently used, and biological reactions and solid–liquid separation can occur in one reactor, reducing the site area and construction cost [3]. AGS can also be used to treat wastewater with various pollutant loads and characteristics.

AGS formation is divided into four stages: cell–cell adhesion, formation of microaggregates, secretion of extracellular polymeric substances (EPS), and maturation of granules [4]. However, the formation of AGS is affected by numerous factors along with strict operating conditions, and its formation is time consuming [5]. In addition, the long granulation period and instability limits the full-scale application of AGS technology. Accordingly,

considerable research has been conducted on new strategies to complement the existing mechanisms of granule formation. Studies on some of the new proposed strategies to improve aerobic granulation have added metal ions such as calcium [6] and granular activated carbon [7]. The addition of exogenous substances can help maintain the stability of the AGS and shorten its granulation period [8].

Recent studies have reported that quorum sensing (QS), a communication system between cells caused by diffusible signaling molecules, is closely related to the floc or biofilm formation process and sludge granulation [9,10]. Bacterial communities regulate their behavior by exchanging chemical signaling molecules to coordinate bacterial responses using QS [11]. The exchange of signaling molecules alters biofilm formation and EPS production. Signaling molecules with appropriate molecular structures can improve microbial activity and enhance granule formation by promoting EPS secretion during microbial attachment [12]. Autoinducers are signaling molecules generated in response to changes in cell density. N-Acyl homoserine lactones (AHLs) are the common autoinducers secreted by Gram-negative bacteria. AHL is abundant in AGS systems and is known to have an appropriate molecular structure for expressing QS. Li et al. [12] reported that the addition of AHL signaling molecules (3-oxo-C6-HSL and C6-HSL) increased biomass growth rate, microbial activity, and nitrification biofilm formation. The addition of four exogenous AHLs (C6-HSL, 3OC6-HSL, 3OC8-HSL, and 3OC12-HSL) with different molecular structures significantly promoted the synthesis of EPS [13]. Exogenous C6-HSL and C8-HSL not only increase nitrogen removal efficiency but also have important effects on the production of EPS and the microbial community [14]. Ma et al. [15] tested granulation formation using different AHL types and found that exogenous C10-HSL and C8-HSL promoted anaerobic granulation.

In general, the process of biofilm formation consists of five steps: initial attachment, irreversible attachment by secretion of EPS, proliferation, maturation, and dispersal. Exogenous AHL can accelerate the initial granulation process [10]. Fang et al. [16] reported that the exogenous addition of AHL accelerated biofilm formation and shortened start-up lag periods by approximately 50%. According to Zhang et al. [17], C8-HSL, 3OHC8-HSL, and 3OHC12-HSL can increase tryptophan and protein-like substances, zeta potential, and average granule size. Therefore, it was argued that it may be the main AHL component regulating the microbiome in AGS and that the exogenous addition of AHL is required to enhance the formation of AGS. The addition of exogenous AHL can have a positive effect on granulation, but the effect of AHL signaling molecules on the granulation process is not fully understood [14].

In this study, the effect of AHL-mediated QS on aerobic granulation on sludge characteristics and the granulation process was confirmed by adding several types of AHL in a batch method. This was to determine the most suitable AHL type for the AGS system. The sludge characteristics, particle size, EPS, and biofilm formation were investigated.

## 2. Materials and Methods

### 2.1. Batch Reactor Operation

Batch experiments were performed to investigate the granulation process and the type of AHL signaling molecule most suitable for the AGS system. Activated sludge (300 mL) and synthetic wastewater (300 mL) were mixed in a 1 L reactor (Figure 1). One of the five reactors, without the addition of AHL, was used as a control. The remaining four reactors were injected with 5 μM N-Hexanoyl-L-homoserine lactone (C6-HSL), N-Octanoyl-L-homoserine lactone (C8-HSL), N-Decanoyl-L-homoserine lactone (C10-HSL), or N-Dodecanoyl-L -homoserine lactone (C12-HSL) (Figure 1). Sludge returned from the secondary clarifier of the Jungnang Sewage Treatment Plant in Seoul was used as the seeding sludge. The TSS and VSS of the sludge were 3260 mg/L and 2658 mg/L, respectively. Synthetic wastewater was prepared by simulating the side streams of a sewage treatment plant [18]. The composition was as follows: COD 2480 mg/L (sodium acetate), T-N 86.03 mg/L (NH$_4$Cl), T-P 24.57 mg/L (K$_2$HPO$_4$ and KH$_2$PO$_4$). The reactor

was mixed at 90 rpm using an agitator, and the water temperature was adjusted to be maintained at 25 ± 2 °C. Air was supplied by an air diffuser at a rate of 200 mL/min, and aerobic conditions were maintained.

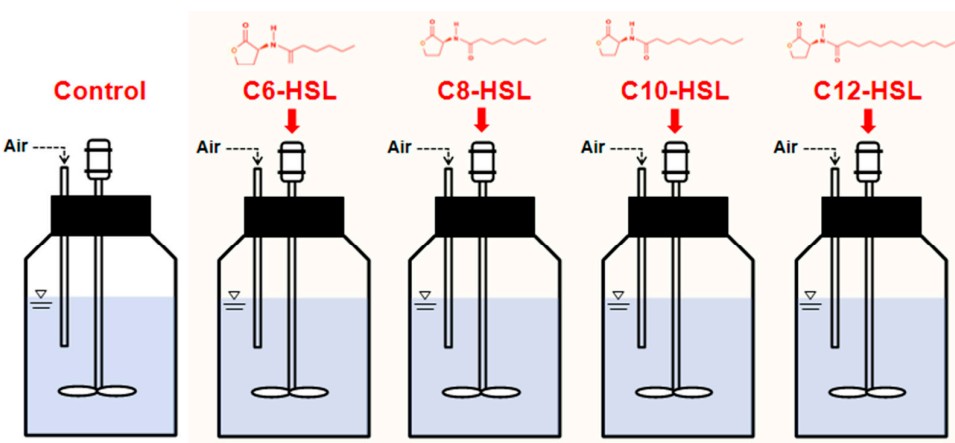

**Figure 1.** Schematic diagram of the batch reactors indicating the N-acyl-homoserine lactone added to each.

## 2.2. Analytical Methods

### 2.2.1. Sludge Characteristic Test

Batch reactor parameters such as sludge volume index (SVI), mixed liquor suspended solids (MLSS), and mixed liquor volatile suspended solids (MLVSS) were analyzed according to standard methods (23rd Edition). Optical microscopy (BX51, Olympus Co., Tokyo, Japan) and field-emission scanning electron microscopy (FE-SEM, SU8010, Hitachi Ltd., Tokyo, Japan) were used to examine the morphology of the sludge. The particle size distribution of the granules was analyzed using a particle size analyzer (Malvern Mastersizer 2000, Malvern Panalytical Ltd., Malvern, UK) [19].

### 2.2.2. EPS Analysis

For EPS extraction, the heat extraction method, which is the most effective extraction method, was used [20], and a modified version of the existing heat extraction method was used [21,22]. The sludge (25 mL) in the reactor was sampled and centrifuged at $4000 \times g$ for 5 min, the supernatant was discarded, and the remaining solids were resuspended in 70 °C phosphate-buffered saline solution and mixed for 1 min, and then loosely bound EPS (LB-EPS) was measured in the supernatant after centrifugation ($4000 \times g$, 10 min). After extraction, the remaining solids were resuspended in 60 °C phosphate-buffered saline solution, mixed for 1 min, and reacted for 30 min in a water bath with a water temperature of 60 °C. To measure TB-EPS (tightly bound EPS), the supernatant was centrifuged at $4000 \times g$ for 15 min and filtered through a 0.45 μm filter. The extracted EPS was quantified as the sum of its main components, protein (PN) and polysaccharide (PS). Proteins were analyzed using a Bio-Rad protein assay (Protein Assay Kit I 5000001, Bio-Rad, Boulder, CO, USA) based on the Bradford dye-binding method [23,24]. For protein quantification, a standard curve was prepared using 2 mg/mL bovine serum albumin stock solution. After placing 0.5 mL of the sample and 0.5 mL protein dye (Bio-Rad, Boulder, CO, USA) in a micro cuvette, it reacted for 15 min, and then the absorbance was measured at a wavelength of 595 nm using a UV/VIS spectrophometer (OPTIZEN POP, Mecasys Co., Ltd., Daejeon, Republic of Korea). PS was measured using a TOC analyzer (Multi N/C-3100, Analytik Jena AG, Jena, Germany).

The extracted EPS solution was analyzed using a fluorescence excitation–emission matrix (F-EEM, RF-5301 spectrofluorometer, Shimadzu Co., Kyoto, Japan) to investigate the effect of AHL on organic material composition. The fluorescence characteristics of the

organic materials were scanned using an arc lamp at excitation wavelengths of 220–400 nm (10 nm intervals) and emission wavelengths of 280–600 nm (1 nm intervals).

### 2.2.3. Relative Hydrophobicity

The relative hydrophobicity (RH) was measured by partially modifying the hydrocarbon-hexane extraction method [25]. In a separatory funnel, add the same amount of n-hexane as the 15 mL sample, mix for 10 min, and leave for 30 min to completely separate the two phases; the aqueous phase was then transferred, and the absorbance was measured at 600 nm. RH (%) was calculated using the absorbance before and after n-hexane treatment, as shown in Equation (1).

$$RH\ (\%) = (1 - A_0/A) \times 100 \tag{1}$$

where $A_0$ is the $OD_{600}$ before treatment with n-hexane, and A is the $OD_{600}$ after treatment with n-hexane.

### 2.2.4. Biofilm Analysis

To analyze biofilm formation and characteristics, the membrane was cut to a size of 150 × 150 mm and fixed in a batch reactor and operated for 24 h together with sludge. The membrane pieces to which the microorganisms were attached were washed twice with 0.9% NaCl solution and stained with the Molecular Probes (LIVE/DEAD BacLight Bacterial Viability Kit, Eugene, OR, USA) containing SYTO9 and propidium iodide (PI). SYTO9 is a green fluorescent dye that can penetrate both living and dead cells, whereas PI is a red fluorescent dye that can only penetrate dying or dead cells with damaged cell membranes [26]. For biofilm staining, 3 μL SYTO9 and 3 μL PI solution were mixed with 1 mL distilled water, and 200 μL of the mixture was pipetted to cover the entire membrane surface. The Petri dish containing the membrane was covered with aluminum foil, left in the dark for 30 min, and then carefully washed with distilled water to remove excess stain [27]. Confocal laser scanning microscopy (CLSM, LSM 810, Carl Zeiss, Oberkochen, Germany) was used to investigate the thickness, shape, and distribution of the biofilm attached to the membrane surface and to obtain images. All images were analyzed using an image analysis program (Zen software, Carl Zeiss Co., Oberkochen, Germany), and the CLSM images of the biofilms formed on the membrane surface were quantified using COMSTAT image analysis software [28].

## 3. Results and Discussion

### 3.1. Effect of AHL on the Expression of EPS

#### 3.1.1. Content of EPS

In general, EPS consists of two layers, TB-EPS and LB-EPS, based on the relationship between the EPS and cells. The TB-EPS forms an inner layer surrounding the microbial cells, and the LB-EPS forms an outer layer covering the TB-EPS. The composition of EPS has different effects on sludge properties, and the addition of exogenous AHL not only changes the distribution of EPS but also affects the functional groups [29].

Different types of AHL signaling molecules have slightly different effects on EPS content. TB-EPS was 18.49 mg/g VSS and 18.68 mg/g VSS in the reactors added with C8-HSL and C12-HSL, respectively. It showed a higher compared to the control group (17.91 mg/g VSS), unlike C6-HSL and C10-HSL (Figure 2). Several studies have reported that TB-EPS maintains granule formation and structural stability [30,31]. LB-EPS significantly increased compared to TB-EPS, and all reactors to which AHL signal molecules were added showed increased LB-EPS compared to the control group. This may be because the shear resistance of the LB-EPS is lower than that of the TB-EPS [32]. Overall, C8-HSL was most effective at increasing the EPS content. Among the AHL types, TB-EPS and LB-EPS in the reactor with C8-HSL added were 18.49 and 74.07 mg/g VSS, respectively, which represented increases of 3.15% and 53.76% compared to the control group. This is in agreement with Yeon et al. [33], who asserted that the addition of C8-HSL increases the EPS content.

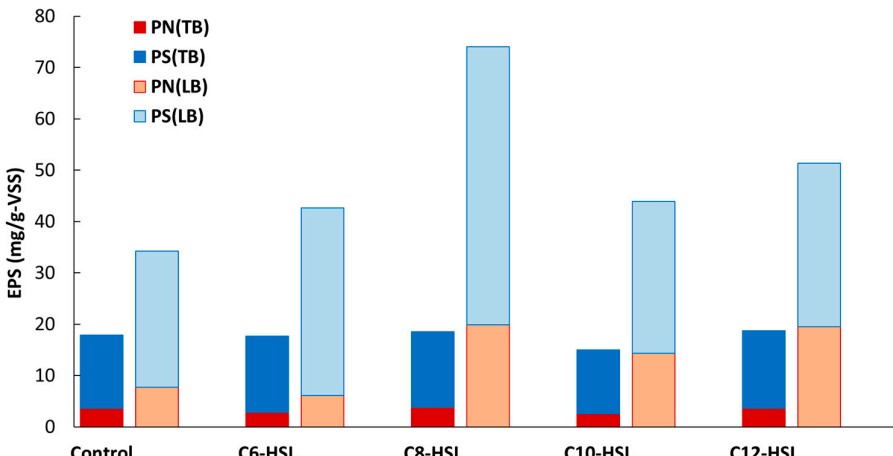

**Figure 2.** Changes in EPS content according to AHL type. PN = protein; PS = polysaccharide; LB = loosely bound; TB = tightly bound.

EPS can affect sludge flocculation and granulation mainly by changing surface properties such as surface charge and hydrophobicity. Among its constituents, PN plays an important role in crosslinking because it has a better affinity for cations than other constituents [34]. The synthesis of another component, PS, influences the cell attachment capacity and may aid in the formation and stability of microbial aggregates [15]. In this study, although LB-PS accounted for a larger proportion of the total EPS, the addition of AHL seemed to have a greater effect on the increase in LB-PN, with a maximum of 2.41 times for LB-PN and 2.05 times for LB-PS. The increase in EPS was found to be significantly influenced by AHL signaling molecules, with polysaccharides playing a particularly noteworthy role. AHL signaling molecules have been reported to affect the early stages of granule formation, leading to increases in biomass concentration, hydrophobicity, and sedimentation [35].

3.1.2. Composition of EPS

F-EEM was analyzed to investigate the effect of adding AHL signaling molecules on the composition of organic materials in EPS. In the reactor with C8-HSL, peaks T1 and T2, which corresponded to protein-like substances among the LB-EPS constituents, were the highest (Figure 3). Unlike the other AHL additions, peaks B1 and B2 also appeared to be prominent. This result corresponds to the protein content of the EPS (Figure 2). AHL signaling molecules promoted the secretion of protein-like substances, and the addition of C8-HSL had the most significant effect. As shown in Figure 3b,c, peaks T1 and T2 in TB-EPS were highest in the reactor with C8-HSL. In the reactors with the addition of C8-HSL, C10-HSL, and C12-HSL, the fluorescent strengths of peaks A and C were high, which corresponded to fulvic acid and humic acid-like substances in EPS (Figure 4). Zhu et al. [34] reported that protein-like substances are advantageous for the formation and stability of granulated sludge, whereas humic and fulvic acid-like substances have the potential to cause deterioration of granulation. In addition, humic acid-like organics are mainly produced by cell death and degradation of organisms, and their accumulation may not be beneficial for the aggregation of microorganisms [36].

The addition of AHL signaling molecules clearly changed the composition of EPS and could have a positive effect on granulation. In this study, we found that C8-HSL had the greatest effect on both types of EPS containing tryptophan and aromatic protein-like substances. This is consistent with the results of previous studies showing that tryptophan and aromatic protein-like substances are the main substances in the formation of EPS in early sludge, as short-chain AHL can be more advantageous for the formation of granules than long-chain AHL [37]. This suggests that the addition of C8-HSL may contribute to granule formation.

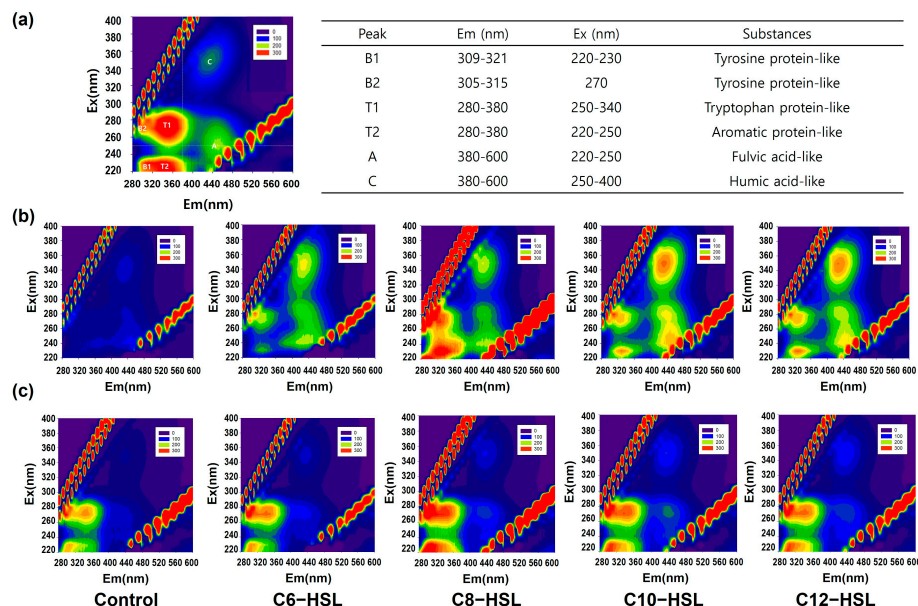

**Figure 3.** F-EEM spectra (**a**) an example (**b**) LB-EPS and (**c**) TB-EPS.

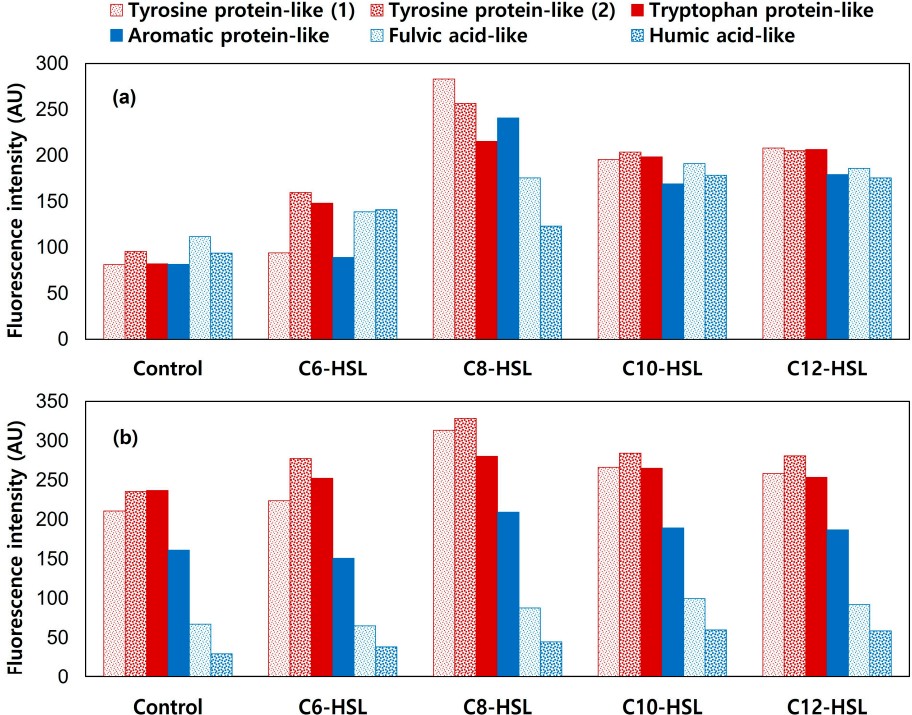

**Figure 4.** Comparison of fluorescence intensities in the four AHL (**a**) LB-EPS and (**b**) TB-EPS.

## 3.2. Effect of AHLs on the Relative Hydrophobicity and Biofilm Formation

### 3.2.1. Change in Relative Hydrophobicity of Sludge

Relative hydrophobicity is one of the main properties of the biomass that forms biofilms. Increased cell-surface hydrophobicity promotes cell-to-cell interactions and is considered a force that causes granulation [38]. The addition of AHL signaling molecules increased the RH values of the sludge flocs, regardless of the type (Figure 5). The AHL showing the highest increase in hydrophobicity was C8-HSL, with an RH value of 16.36% in the control reactor and an RH value of 25.00%, which increased approximately 1.53 times in the reactor containing C8-HSL. The type of AHL that had the smallest effect on the increase

in the RH value was C10-HSL, which showed an RH value of 20.00%, approximately 1.22-fold higher than that of the control reactor without AHL input.

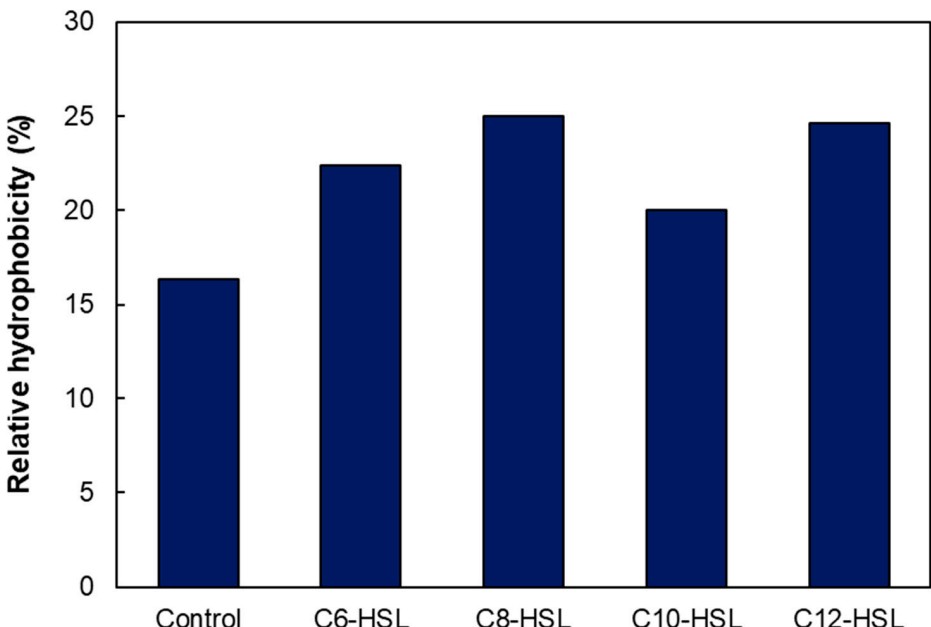

**Figure 5.** Effect of AHL on relative hydrophobicity of sludge.

As the hydrophobicity of the sludge floc increases, bioflocculation increases, and adhesion can be improved [25]. Thus, the higher the RH value, the better the aggregation properties; the relatively low surface charge and high hydrophobicity promote the aggregation of microorganisms [39]. Figure 5 shows that the sludge floes formed when AHL is applied have excellent hydrophobicity. This increase in hydrophobicity may be a factor in forming granular sludge with larger sizes and denser structures.

### 3.2.2. Biofilm Formation

The biofilm was observed using CLSM image analysis, and information on its structure, morphology, and biomass was obtained. Because the structure of the biofilm was not homogeneously distributed, both the central portions of the membrane were analyzed. Figure 6 shows the CLSM image of the biofilm above the membrane; the stronger the fluorescence of microorganisms, the brighter the green fluorescence appears [24]. Most cells appear to have green fluorescence, which indicates a living cell owing to its high metabolism. However, the control reactor to which AHL was not added was rarely observed in the image because the biofilm was not well formed.

The effects of AHL on biofilm formation were compared by quantifying the volume and thickness of the biofilm using CLSM images. The biomass volume in the membrane operated for 24 h was $7.82 \pm 6.17$ $\mu m^3/\mu m^2$ and $6.99 \pm 10.22$ $\mu m^3/\mu m^2$ in C8-HSL and C12-HSL, respectively, which was larger than $3.58 \pm 8.72$ $\mu m^3/\mu m^2$ in the control reactor. The average biomass thickness was also the highest in the reactor with C8-HSL at $15.31 \pm 12.96$ $\mu m$, followed by the reactor with C12-HSL at $14.45 \pm 10.69$ $\mu m$. These results suggest that the addition of AHL increased the thickness of the biofilm and biomass. This may be because AHLs promote EPS secretion, which improves bacterial attachment and facilitates the construction of 3D biofilms [40]. The addition of QS-related substances, such as AHLs, can contribute to significant changes in EPS and biofilms, which can be achieved through QS signaling [13,41]. Thus, the addition of AHL can promote biofilm formation, increase adhesion at the beginning of granule formation, and have a positive effect on AGS.

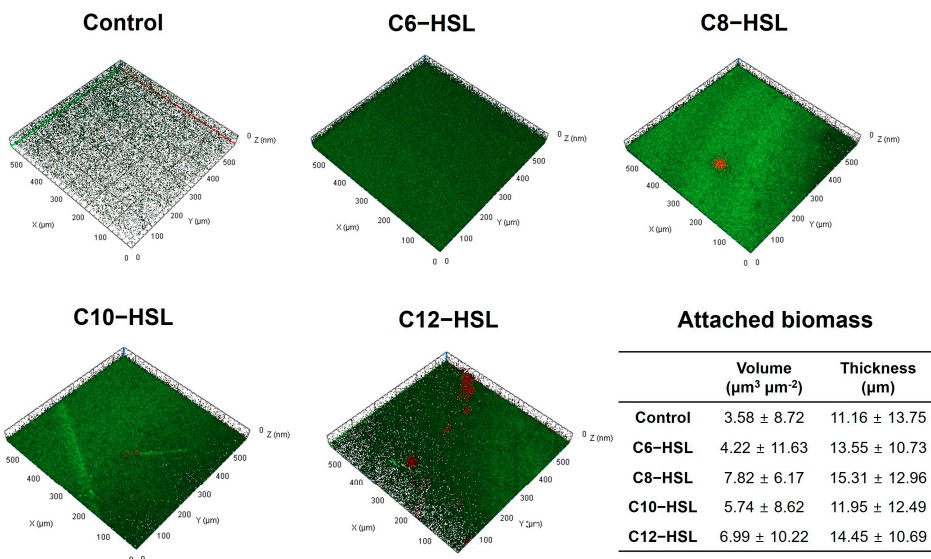

**Figure 6.** CLSM images of the membrane surface and biofilm characteristics.

### 3.3. Formation of Granules

### 3.3.1. Change in the Size of Sludge Floc with the Addition of AHL

The addition of AHL signaling molecules increased the size of the granules; however, the difference between the types was not significant. The largest granules were formed in reactors with C6-HSL and C8-HSL (Figure 7). Zhang et al. [42] showed that the addition of exogenous AHL signaling molecules in the early stage of granulation can promote microbial activity and biomass growth rate and improve sludge granulation. In addition, among the AHLs, C8-HSL and C10-HSL have a strong positive correlation with average granule size, and there is a strong correlation between AHL levels and granule formation [15].

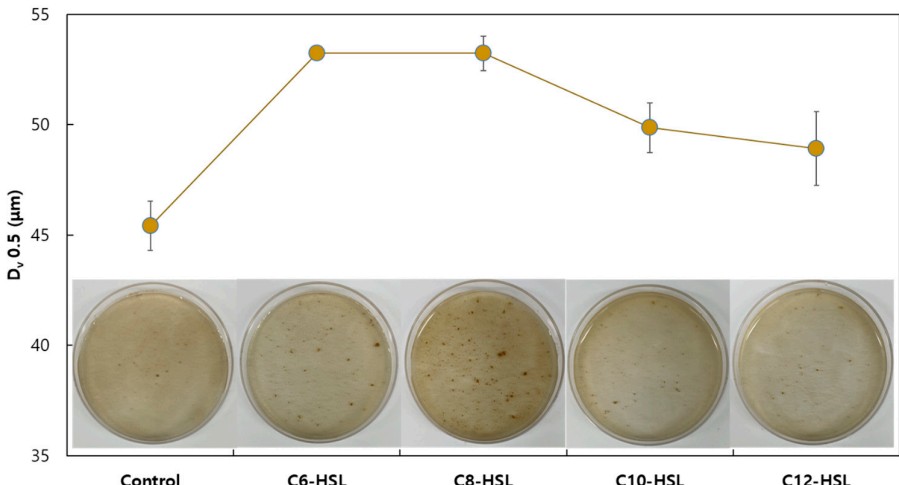

**Figure 7.** Particle size distribution and corresponding sludge image.

### 3.3.2. MLSS and Settleability

The effect of the AHL type on MLSS and MLVSS was significant (Figure 8). Three weeks postoperatively, the MLSS in the control group without AHL was 2125 mg/L, whereas when C8-HSL was added, the AHL signal molecules increased to 2930 mg/L. Furthermore, the addition of AHL resulted in a VS/TS ratio of over 85%, irrespective of the type, surpassing that of the control group. This finding corroborates previous research demonstrating that AHL can impact biomass sedimentation and biological activity during granulation [2]. Figure 8 shows the SVI results indicating sludge settling properties. In

general, the settling properties decrease when the VS/TS ratio increases; however, in this study, the settling properties of the sludge improved despite the increase in the VS/TS ratio due to the addition of AHL signal molecules. It is believed that this is attributable to a rise in EPS content and, in particular, an increase in PS, which contributes to the formation and stability of microbial aggregates, despite the increase in the VS/TS ratio. EPS affects the electronegativity of the sludge surface, PN has a negative charge, and PS has a positive charge. Therefore, the PS/PN ratio can determine the surface charge of EPS, and as the PS/PN ratio increases, the settleability of the sludge can be improved because the electronegativity decreases. In addition, sludge granulation can be promoted because the binding force between bacteria and water molecules is reduced [43]. This is consistent with the result that settleability is the best in the C8-HSL condition with the highest PS/PN ratio. These results demonstrate that AHL signaling molecules increase biomass concentration and improve precipitability [35].

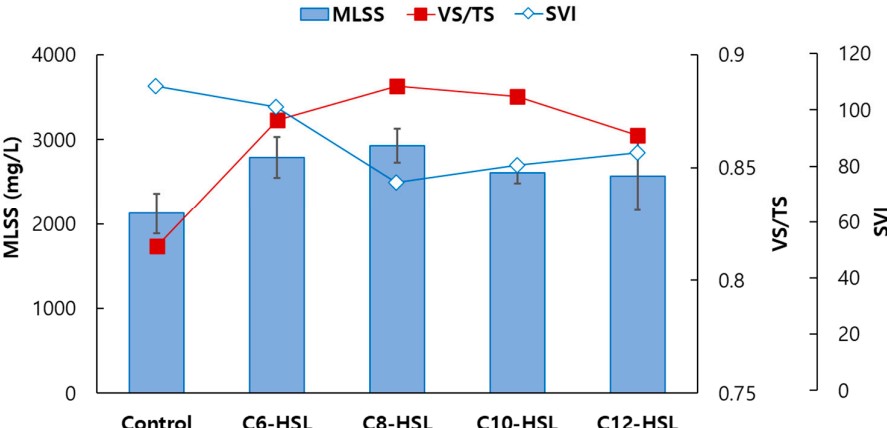

**Figure 8.** Changes in MLSS, VS/TS, and SVI according to the type of AHL added.

## 4. Conclusions

In this study, the effects of AHL signaling molecules with appropriate structures on sludge properties and aerobic granulation were investigated. It was found that the secretion of EPS was promoted by the addition of AHL signaling molecules. The application of AHL changed the characteristics of the sludge regardless of the type of AHL added, increased the amount of biomass and relative hydrophobicity, and improved the settling properties of the sludge. In the CLSM analysis, the biofilm showed brighter green fluorescence due to living cells, and it was possible to determine the difference according to the type of AHL in the formation of the biofilm. The addition of AHL signaling molecules clearly changed the composition of EPS and promoted the secretion of protein-like substances, with the addition of C8-HSL having the greatest impact. Among the exogenous signaling molecules, C8-HSL appears to be the most suitable for promoting aerobic granulation by significantly increasing the EPS content, biofilm thickness, and relative hydrophobicity. When C8-HSL was added, LB-EPS was considered to be the most effective AHL signaling molecule in terms of sludge property change and granulation, with 74.07 mg/g VSS, a 53.76% increase compared to the control. The exogenous addition of AHL can be an effective strategy because it induces EPS production and enhances biofilm formation, thereby shortening the granulation period and improving the stability of the AGS system.

**Author Contributions:** Conceptualization, H.C.; Methodology, E.L.; Validation, K.J.M.; Formal analysis, K.J.M.; Investigation, E.J.; Data curation, E.J.; Writing—original draft preparation, E.J.; Writing—review and editing, K.J.M. and K.Y.P.; Visualization, E.L.; Supervision, K.Y.P.; Project administration, H.C. All authors have read and agreed to the published version of the manuscript.

**Funding:** This research was funded by Korea Ministry of Environment grant number 2021002690005.

**Data Availability Statement:** Data available on request.

**Acknowledgments:** This work was supported by the Korea Environment Industry & Technology Institute (KEITI) through a project for developing innovative drinking water and wastewater technologies funded by the Korea Ministry of Environment (MOE) (2021002690005).

**Conflicts of Interest:** The authors declare no conflict of interest.

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
