# Peer review of "Acceleration of Aerobic Granulation in Sidestream Treatment with Exogenous Autoinducer"

_water, doi:10.3390/w15122173_

Round 1

Reviewer 1 Report

Page 1, line 17 Missing punctuation.

Page 2, lines 57-59 There are absences of some contact information about EPS in the above literature.

Page 3, line 117 mixed liquor suspended solids (MLVSS)?

Page 3, line 118 Missing the citation of the standard method.

Page 4, line 189 This sentence seems somewhat redundant if it does not introduce another theory of EPS layers.

Page5, line 198 (Bhatia et al., 2103), 2103?

The serial number of the entire article title is incorrect.

Figure 4 should be on a separate page. And why use two similar “Tyrosine protein-like” in Figure 4 a)?

Page 6, lines 255-257 It could not get the conclusion about “the application of AHL could form granulated sludge with a larger size and denser structure” only through Figure 5.

Page 6, lines 268-271, and Figure 6 Some standard deviations are greater than the average, hence, it is hard to get a plausible conclusion.

Page 6, line 285 How can draw conclusions that the difference between types was not significant without statistical analysis?

Page 7, MLSS and settle ability Although the SVI really decreased in other groups which add AHL, it is the result of AHL promoting the granulation process. There might due to more granulation that led to a lower SVI than the control group rather than AHL improved the precipitability. So, it needs more evidence to prove it.

Page 7, line 341 “VS/ST”?

Reviewer 2 Report

The contents were worth to study and some useful conclusions were obtained. Reviewer recommends the publication of this paper in this journal. However, before acceptance, the following major concerns must be addressed:

1. Please reorganize the language of the Abstract, the first half is illogical and the data displayed in the abstract should be simplification.

2. The language and the grammar of the article need to be optimized.

3. Authors showed biofilm formation from CLSM images, however cell morphology should be shown in detail from SEM and 2D CLSM for a comparison.

4. Biofilm surface can be shown in AFM and also AHL/biofilm fluorescence intensities could have seen from flow cytometer  

4. Similarly, the author should improve the discussion part by providing more information on these particularly species rather reporting others report. Author should discuss more rather than writing observation. What is the real motive of this work?

There are some sentences with grammar mistakes and missing articles throughout the manuscript. Some sentences need to be reworded. 
